# Floquet-enhanced spin swaps

Haifeng Qiao [1], Yadav P. Kandel [1], John S. Van Dyke [2], Saeed Fallahi[3,4], Geoffrey C. Gardner[4,5], Michael J. Manfra [3,4,5,6], Edwin Barnes[2] & John M. Nichol [1]✉

The transfer of information between quantum systems is essential for quantum communication and computation. In quantum computers, high connectivity between qubits can improve the efficiency of algorithms, assist in error correction, and enable high-fidelity readout. However, as with all quantum gates, operations to transfer information between qubits can suffer from errors associated with spurious interactions and disorder between qubits, among other things. Here, we harness interactions and disorder between qubits to improve a swap operation for spin eigenstates in semiconductor gate-defined quantum-dot spins. We use a system of four electron spins, which we configure as two exchange-coupled singlet–triplet qubits. Our approach, which relies on the physics underlying discrete time crystals, enhances the quality factor of spin-eigenstate swaps by up to an order of magnitude. Our results show how interactions and disorder in multi-qubit systems can stabilize non-trivial quantum operations and suggest potential uses for non-equilibrium quantum phenomena, like time crystals, in quantum information processing applications. Our results also confirm the long-predicted emergence of effective Ising interactions between exchange-coupled singlet–triplet qubits.

[1] Department of Physics and Astronomy, University of Rochester, Rochester, NY, USA. [2] Department of Physics, Virginia Tech, Blacksburg, VA, USA. [3] Department of Physics and Astronomy, Purdue University, West Lafayette, IN, USA. [4] Birck Nanotechnology Center, Purdue University, West Lafayette, IN, USA. [5] School of Materials Engineering, Purdue University, West Lafayette, IN, USA. [6] School of Electrical and Computer Engineering, Purdue University, West Lafayette, IN, USA. ✉email: john.nichol@rochester.edu

Over the past decades, quantum information processors have undergone remarkable progress, culminating in recent demonstrations of their astonishing power[1]. As quantum information processors continue to scale-up in size and complexity, new challenges come to light. In particular, maintaining the performance of individual qubits and high connectivity are both essential for continued improvement in large systems[2].

At the same time, developments in nonequilibrium many-body physics have yielded insights into many-qubit phenomena, which feature, in some sense, improved performance of many-body quantum systems when disorder and interactions are included. Chief among these phenomena are many-body localization[3] and time crystals[4–8]. Although these phenomena are interesting in their own right, applications of these concepts are only beginning to emerge.

In this work, we exploit discrete-time-crystal (DTC) physics to demonstrate Floquet-enhanced spin-eigenstate swaps in a system of four quantum dot electron spins. When we harness interactions and disorder in our system, the quality factor of spin-eigenstate swaps improves by nearly an order of magnitude. As we discuss in detail further below, this system of four exchange-coupled single spins undergoing repeated SWAP pulses maps onto a system of two Ising-coupled singlet–triplet (ST) qubits undergoing repeated $\pi$ pulses. Periodically driven Ising-coupled spin chains are the prototypical example of a system predicted to exhibit DTC behavior[4]. Experimental signatures of DTC behavior have been observed in many systems[9–12], but nearest-neighbor Ising-coupled spin chains have yet to be experimentally investigated in this regard.

Our system of two ST qubits is clearly not a DTC in the strict sense, because it is not a many-body system[13]. However, this system does exhibit some of the key characteristics of DTC behavior, including robustness against interactions, noise, and pulse imperfections[13,14]. We also find that the required experimental conditions for observing the quality-factor enhancement are identical to some of the theoretical conditions for the DTC phase in infinite spin chains. In total, these observations suggest the Floquet-enhanced spin-eigenstate swaps in our device are closely related to discrete time-translation symmetry breaking.

Our results also illustrate how nonequilibrium many-body phenomena could potentially be used for quantum information processing. On the one hand, we observe Floquet-enhanced $\pi$ rotations in two ST qubits. But on the other hand, these ST $\pi$ rotations correspond to spin-eigenstate swaps, when we view the system as four single spins. The enhanced spin-eigenstate swaps are not coherent SWAP gates, but instead are "projection-SWAP" gates[15]. Because of the critical importance of such operations for reading out linear qubit arrays, these results may point the way toward the use of nonequilibrium quantum phenomena in quantum information processing applications, especially for initialization, readout, and information transfer. Moreover, recent theoretical work shows how entangled states can be preserved, and robust single-, and two-qubit gates can be implemented, within this framework[16]. Our results are also significant because they provide experimental evidence of the predicted Ising coupling that emerges between exchange-coupled ST qubits[17].

## Results

### Device and Hamiltonian.
We fabricate a quadruple quantum dot array in a GaAs/AlGaAs heterostructure with overlapping gates (Fig. 1a)[18–20]. The confinement potentials of the dots are controlled through "virtual gates"[21–24]. Two extra quantum dots are placed nearby and serve as fast charge sensors[25,26]. We configure the four-spin array into two pairs ("left" and "right") for initialization and

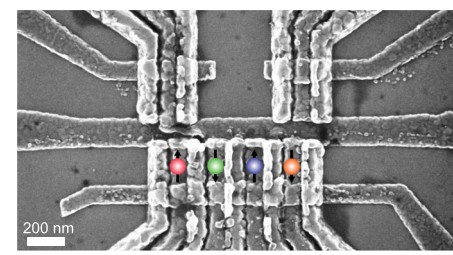

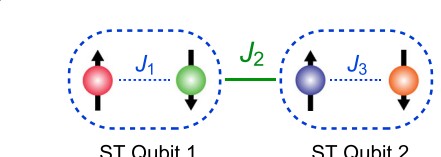

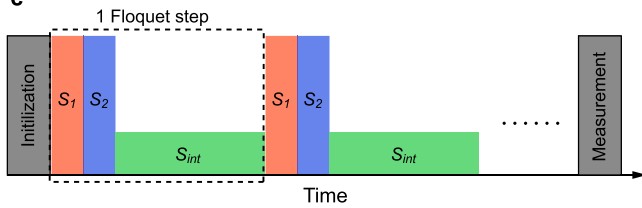

**Fig. 1 Experimental setup. a** Scanning electron micrograph of the quadruple quantum dot device. The locations of the electron spins are overlaid. **b** Schematic showing the two-qubit Ising system in a four-spin Heisenberg chain. **c** The pulse sequence used in the experiments.

readout. Each pair of spins can be prepared in a product state ($|\uparrow\downarrow\rangle$ or $|\downarrow\uparrow\rangle$) via adiabatic separation of a singlet in the hyperfine gradient[27–29]. We can also initialize either pair as $|T_+\rangle = |\uparrow\uparrow\rangle$ by exchanging electrons with the reservoirs[28,30]. Both pairs are measured through spin-to-charge conversion via Pauli spin blockade[27], together with a shelving mechanism[31] for high readout fidelity. Further details about the device can be found in "Methods".

The four-spin array is governed by the following Hamiltonian:

$$H = \frac{h}{4} \sum_{i=1}^{3} J_i(\boldsymbol{\sigma}_i \cdot \boldsymbol{\sigma}_{i+1}) + \frac{h}{2} \sum_{i=1}^{4} B_i^z \sigma_i^z , \quad (1)$$

where $J_i$ is the tunable exchange coupling strength (with units of frequency), $\boldsymbol{\sigma}_i = [\sigma_i^x, \sigma_i^y, \sigma_i^z]$ is the Pauli vector describing the components of spin $i$, $h$ is Planck's constant, and $B_i^z$ is the $z$ component of the magnetic field (also with units of frequency) experienced by spin $i$. $B_i^z$ includes both a large 0.5-T external magnetic field and the smaller hyperfine field. The exchange couplings $J_1$, $J_2$, and $J_3$ are controlled by pulsing virtual barrier gate voltages[32]. We model the dependence of the exchange couplings on the virtual barrier gate voltages in the Heitler–London framework[32,33]. The model allows us to predict the required barrier gate voltages for a set of desired exchange couplings. In our device, we estimate the residual exchange coupling at the idling tuning of the device to be a few MHz.

Heisenberg exchange coupling does not naturally enable the creation of a DTC phase[8]. Additional control pulses can convert the Heisenberg interaction into an Ising interaction[8], which permits the emergence of a DTC phase. A DTC can also be created using a sufficiently strong magnetic field gradient instead of applying extra pulses[34]. Here, we introduce a new method for generating DTC behavior that does not require complicated pulse sequences or large field gradients, but instead relies only on periodic exchange pulses.

To see how we can still obtain an effective Ising interaction in this case, it helps to view each pair of spins as an individual ST

qubit (Fig. 1b)[27]. Specifically, consider the scenario where the joint spin state of each pair is confined to the subspace spanned by $|S\rangle = \frac{1}{\sqrt{2}}(|\uparrow\downarrow\rangle - |\downarrow\uparrow\rangle)$ and $|T_0\rangle = \frac{1}{\sqrt{2}}(|\uparrow\downarrow\rangle + |\downarrow\uparrow\rangle)$. According to ref. [17], when $J_1 = J_3 = 0$ and $J_2 > 0$, the effective Hamiltonian of the system is

$$H_{\text{eff}} = \frac{h}{2}(\Delta_{12} + \bar{B})\tilde{\sigma}_1^z + \frac{h}{2}(\Delta_{34} + \bar{B})\tilde{\sigma}_2^z - \frac{h}{4}J_2\tilde{\sigma}_1^z\tilde{\sigma}_2^z . \quad (2)$$

Here, $\tilde{\sigma}_k^z$ is the Pauli $z$-operator for ST qubit $k$, $\Delta_{ij} = B_i^z - B_j^z$ is the intraqubit gradient between spins $i$ and $j$, and $\bar{B}$ is the effective global magnetic field gradient, which depends on $\Delta_{ij}$ and $J_i$ (ref. [17]). In this system, all magnetic gradients result from the hyperfine interaction between the electron and nuclear spins[35]. The gradients are quasistatic on typical qubit manipulation timescales[36]. The basis for the ST-qubit operator $\tilde{\sigma}^z$ is $\{|\uparrow\downarrow\rangle, |\downarrow\uparrow\rangle\}$ provided that $J_2 \ll |B_2 - B_3|$ (refs. [17,36,37]). In our experiments, the typical value of $J_2$ is a few MHz, while the typical value of the magnetic field gradient in the device is tens of MHz. Now let us define

$$S_{\text{int}} = \exp\left[-\frac{i}{\hbar}\tau\left(\frac{h}{4}J_2(\boldsymbol{\sigma}_2 \cdot \boldsymbol{\sigma}_3) + \frac{h}{2}\sum_{i=1}^{4}B_i^z\sigma_i^z\right)\right], \quad (3)$$

where $\tau$ is an interaction time. Within the $\{|S\rangle, |T_0\rangle\}$ subspace of each pair, this operator is equivalent to $S_{\text{int}}^{\text{eff}} = \exp\left[-\frac{i}{\hbar}\tau H_{\text{eff}}\right]$, and it describes the evolution of the two Ising-coupled qubits[17]. Systems of exchange-coupled ST qubits have been the focus of significant theoretical research[17,38–40]. Until now, such a system has evaded implementation.

In the case when $J_2 = 0$, but when $J_1, J_3 > 0$, the overall Hamiltonian describes two uncoupled ST qubits. Thus, let us define

$$S_1 = \exp\left[-\frac{i}{\hbar}t_1\left(\frac{h}{4}J_1(\boldsymbol{\sigma}_1 \cdot \boldsymbol{\sigma}_2) + \frac{h}{2}\sum_{i=1}^{4}B_i^z\sigma_i^z\right)\right], \quad (4)$$

$$S_2 = \exp\left[-\frac{i}{\hbar}t_2\left(\frac{h}{4}J_3(\boldsymbol{\sigma}_3 \cdot \boldsymbol{\sigma}_4) + \frac{h}{2}\sum_{i=1}^{4}B_i^z\sigma_i^z\right)\right]. \quad (5)$$

In the $\{|S\rangle, |T_0\rangle\}$ subspace of each pair, these operators are equivalent to $S_1^{\text{eff}} = \exp\left[-\frac{i}{\hbar}t_1\frac{h}{2}\left(\Delta_{12}\tilde{\sigma}_1^z + J_1\tilde{\sigma}_1^x\right)\right]$ and $S_2^{\text{eff}} = \exp\left[-\frac{i}{\hbar}t_2\frac{h}{2}\left(\Delta_{34}\tilde{\sigma}_2^z + J_3\tilde{\sigma}_2^x\right)\right]$. In writing $S_1^{\text{eff}}$ and $S_2^{\text{eff}}$, we have ignored overall energy shifts $J_1/4$ and $J_3/4$ of the single-qubit Hamiltonians, because the system dynamics do not depend on these shifts. Assuming $J_1 \gg \Delta_{12}$ and $J_3 \gg \Delta_{34}$, when $t_1J_1 = t_2J_3 = 0.5$, these two operators implement SWAP gates between spins 1–2 and 3–4. Equivalently, they induce nominal $\pi$ pulses about the $x$-axis of each ST qubit. The presence of the intraqubit gradients $\Delta_{12}$ and $\Delta_{34}$ slightly tilts the rotation axis toward the $z$-axis for each ST qubit, introducing uncontrolled errors to the $\pi$ pulses. We can also manually introduce additional pulse errors by changing $J_1$ and $J_3$, while fixing $t_1$ and $t_2$. We represent the error as $\epsilon$, with $J_1 = J_1^\pi(1+\epsilon)$ and $J_3 = J_3^\pi(1+\epsilon)$. Here $J_1^\pi$ and $J_3^\pi$ are the interaction strengths that yield $\pi$ pulses.

**Floquet-enhanced spin swaps**. We define a Floquet operator $U = S_{\text{int}} \cdot S_2 \cdot S_1$ (Fig. 1c), and we repeatedly apply this operator to our system of four spins. As discussed above, $U$ implements spin SWAP gates between spins 1–2 and 3–4 followed by a period of exchange interaction between spins 2 and 3. Equivalently, $U$ implements $\pi$ pulses on both ST qubits and then a period of Ising coupling between them. One might naively imagine that the highest fidelity SWAP operations between spins should occur when $J_2 = 0$ and $\tau = 0$, given the presence of intraqubit hyperfine gradients. In this case, as we have discussed in ref. [29], repeated

SWAP operations are especially susceptible to errors from the hyperfine gradients $\Delta_{ij}$.

However, by allowing $J_2 > 0$ and $\tau > 0$, we find specific conditions in which we observe a significant enhancement of the spin-eigenstate-swap quality factor (Fig. 2). To explore this phenomenon, we prepare each ST qubit in $|\uparrow\downarrow\rangle$ or $|\downarrow\uparrow\rangle$. (The specific state is governed by the sign of $\Delta_{12}$ and $\Delta_{34}$, which are random quasistatic gradients resulting from the nuclear hyperfine interaction.) We apply multiple instances of the Floquet operator $U$ to the system and measure the ground-state return probabilities for both ST qubits.

First, we set the interaction time $\tau = 1.4\,\mu s$ and SWAP pulse times $t_1 = t_2 = 5$ ns, and apply four Floquet steps. We sweep $J_2$ linearly from 0.05 to 5 MHz (Fig. 2a, b). (Setting $J_2 < 0.05$ MHz would require large negative voltage pulses applied to the barrier gate due to the residual exchange, which could disrupt the tuning of the device.) We also sweep $J_1$ from 80 to 460 MHz, and $J_3$ from 50 to 260 MHz. The ranges of $J_1$ and $J_3$ roughly center around $J_1^\pi$ and $J_3^\pi$, respectively. Away from the center, $J_1$ and $J_3$ induce pulse errors. The experimental values of $t_1J_1^\pi$ and $t_2J_3^\pi$ are much larger than 0.5, because the voltage pulses experienced by the qubits have rise times of ~1 ns (see "Methods" and Supplementary Fig. 1). To compensate for the pulse rise times (which are slightly different for each qubit), $t_1J_1^\pi$ and $t_2J_3^\pi$ must be larger than 0.5 in order to properly induce $\pi$ pulses.

Clear, bright diamond patterns are visible in the data (Fig. 2a, b). These bright regions correspond to improved spin-eigenstate-swap quality factors. Note that the brightest regions correspond to configurations when $J_2 > 0$. Note also that the diamonds are approximately periodic in $J_2\tau$, as expected for a Floquet operator. We repeat the same experiments with $\tau = 1\mu s$, and we observe similar diamond patterns, although they have an increased period in $J_2$ (Fig. 2e, f). The diamond patterns of ST qubit 2 appear narrower due to the large hyperfine gradient $\Delta_{34}$, which causes larger pulse errors and reduces the size of the quality-factor-enhancement region. These data from an effective two-qubit system resemble predicted DTC phase diagrams of a true nearest-neighbor many-body system (see "Methods" and Supplementary Figs. 2 and 3)[7,8].

Our simulations agree well with the data (Fig. 2c, d, g, h; see "Methods"). In the simulations, the diamond pattern is periodic in $J_2\tau$ with the periodicity of exactly 1, and the strongest quality-factor enhancement occurs at $J_2\tau = 0.5$. In the experimental data, however, the periodicity is slightly larger than 1, and the strongest quality-factor enhancement occurs at $J_2\tau > 0.5$. This is due to the imperfect calibration of the exchange coupling $J_2$ (ref. [32]). In particular, the presence of the hyperfine field gradient makes it difficult to measure and control the exchange couplings with sub-MHz resolution. If our modeling of the exchange coupling were more precise, then we would expect the periodicity of the diamond patterns to be closer to 1 and the quality-factor enhancement to occur closer to $J_2\tau = 0.5$ in the experimental data.

We can interpret our data using a semiclassical model inspired by Choi et al. in ref. [10] to explain DTC behavior (see "Methods"). In brief, an initial state of ST qubit 1, $|\psi_0\rangle = \cos(\theta_0/2)|g\rangle + e^{i\phi_0}\sin(\theta_0/2)|e\rangle$ evolves to $|\psi_f\rangle = e^{-i\phi_1\sigma^z/2}e^{-i\theta\sigma^x/2}e^{-i\phi_2\sigma^z/2}e^{-i\theta\sigma^x/2}|\psi_0\rangle$ after two Floquet steps. Here $\theta \approx \pi$ indicates a nominal $\pi$ pulse, and $\phi_1 = 2\pi(J_2/2 + \Delta_{12} + \bar{B})\tau$, and $\phi_2 = 2\pi(-J_2/2 + \Delta_{12} + \bar{B})\tau$. In this semiclassical model, the effect of ST qubit 2 on ST qubit 1 is to generate the $\pi J_2\tau$ term in the operator that switches sign after each Floquet step, because ST qubit 2 undergoes a nominal $\pi$ pulse. As emphasized in ref. [10], the change in sign of this part between Floquet steps is entirely a result of interactions in the system. The resulting single-qubit rotations in this semiclassical model are reminiscent of

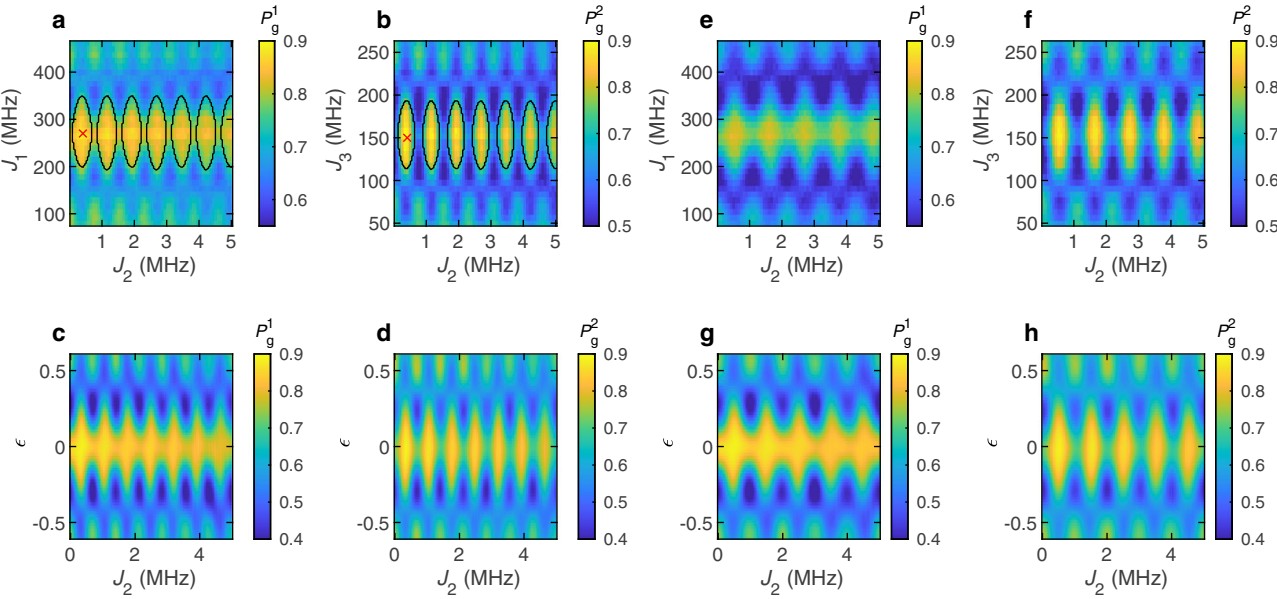

**Fig. 2 Floquet-enhanced π rotations. a**, **b** Measured ground-state return probabilities of (**a**) ST qubit 1 and (**b**) ST qubit 2, after four Floquet steps, with interaction time $\tau = 1.4\,\mu s$. The ranges of $J_1$ and $J_3$ center around $J_1^\pi$ and $J_3^\pi$. The values of $J_1$ and $J_3$ are swept simultaneously. In both figures, the red cross marks the condition for the Floquet-enhanced π rotations. The black ovals are the semiclassical phase boundaries. **c**, **d** Simulated return probabilities of (**c**) ST qubit 1 and (**d**) ST qubit 2, corresponding to the data in (**a**) and (**b**), respectively. **e**, **f** Measured ground-state return probabilities of (**e**) ST qubit 1 and (**f**) ST qubit 2, after four Floquet steps, with interaction time $\tau = 1.0\,\mu s$. $J_1$ and $J_3$ values are the same as in **a** and **b**. **g**, **h** Simulated return probabilities of (**g**) ST qubit 1 and (**h**) ST qubit 2, corresponding to the data in (**e**) and (**f**), respectively. The experimental data in (**a**, **b**, **e**, **f**) are averaged over 8192 realizations. In all figures, $P_g^k$ indicates the ground-state return probability for ST qubit $k$.

dynamical decoupling[10]. We have numerically simulated the semiclassical single-qubit evolution over two Floquet steps for our system (see "Methods"). The black lines in Fig. 2a, b indicate the regions where approximate ST-qubit eigenstates are also exactly eigenstates of the evolution operator over two steps, i.e., they are exactly preseved by the two Floquet steps[10]. The size of these regions confirms that interactions are essential for the effects we observe. Exactly the same enhancement regions are expected for end spins in longer chains, because our system is a nearest-neighbor Ising spin chain. Simulations for an eight-site Ising spin chain at late times show DTC behavior in exactly these regions (see "Methods" and Supplementary Fig. 2).

Next, we also sweep $J_3$ from 220 to 430 MHz. In this case, the range of $J_3$ roughly centers around $J_3^{2\pi}$. The interaction time is $\tau = 1.4\,\mu s$ and the ranges of $J_1$ and $J_2$ remain the same. Again, we apply four Floquet steps and measure the ground-state return probabilities. This time the data do not show diamond patterns (Fig. 3), and the return probability of ST qubit 1 is lower than the Floquet-enhanced return probability shown in Fig. 2a. This indicates that the Floquet enhancement is no longer present. In fact, if either of the Floquet operators $S_1$ or $S_2$ fails to induce approximately a π rotation, then the Floquet enhancement does not appear.

On the one hand, this effect is striking, when one considers the individual spins themselves. Recall that the ST-qubit splittings $\Delta_{12}$ and $\Delta_{34}$ are generated by the hyperfine interaction between the Ga and As nuclei in the semiconductor heterostructure and the electron spins in the quantum dots. Although $\Delta_{12}$ and $\Delta_{34}$ are quasistatic on millisecond timescales, they each independently fluctuate randomly, and can change sign, over the duration of a typical data-taking run, which is ~1 h. Each of the 8192 different realizations for each pixel in the data of Fig. 2 likely contain instances, where both ST qubits have the same or different ground-state spin orientations. (The ground state of each ST

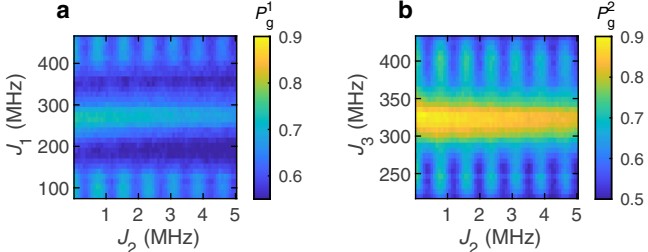

**Fig. 3 Absence of Floquet enhancement due to the omission of a π pulse. a**, **b** Measured ground-state return probabilities of (**a**) ST qubit 1 and (**b**) ST qubit 2, after four Floquet steps, with interaction time $\tau = 1.4\,\mu s$. The ranges of $J_1$ and $J_3$ center around $J_1^\pi$ and $J_3^{2\pi}$, respectively. The values of $J_1$ and $J_3$ are swept simultaneously. The data are averaged over 8192 realizations.

qubit is either $|\uparrow\downarrow\rangle$ or $|\downarrow\uparrow\rangle$, depending on the sign of the instantaneous hyperfine gradient.)

Thus, the data of Fig. 2 likely include realizations with all possible combinations of the orientations of spins 2 and 3 before the interaction period. Despite the random orientations of spins 2 and 3, the Floquet enhancement still appears. It might therefore seem that whether or not spins 1–2 or 3–4 undergo a SWAP before the interaction period should not affect the behavior of the system. However, as shown in Fig. 3, implementing a 2π rotation, as opposed to a π rotation, on one of the ST qubits eliminates the Floquet enhancement.

On the other hand, when one considers the semiclassical picture described above, the absence of a π pulse on one of the ST qubits spoils the semiclassical decoupling evolution discussed above and in ref. [10]. In this case, ST-qubit eigenstates are no longer eigenstates of two instances of the Floquet operator, and the enhancement no longer occurs.

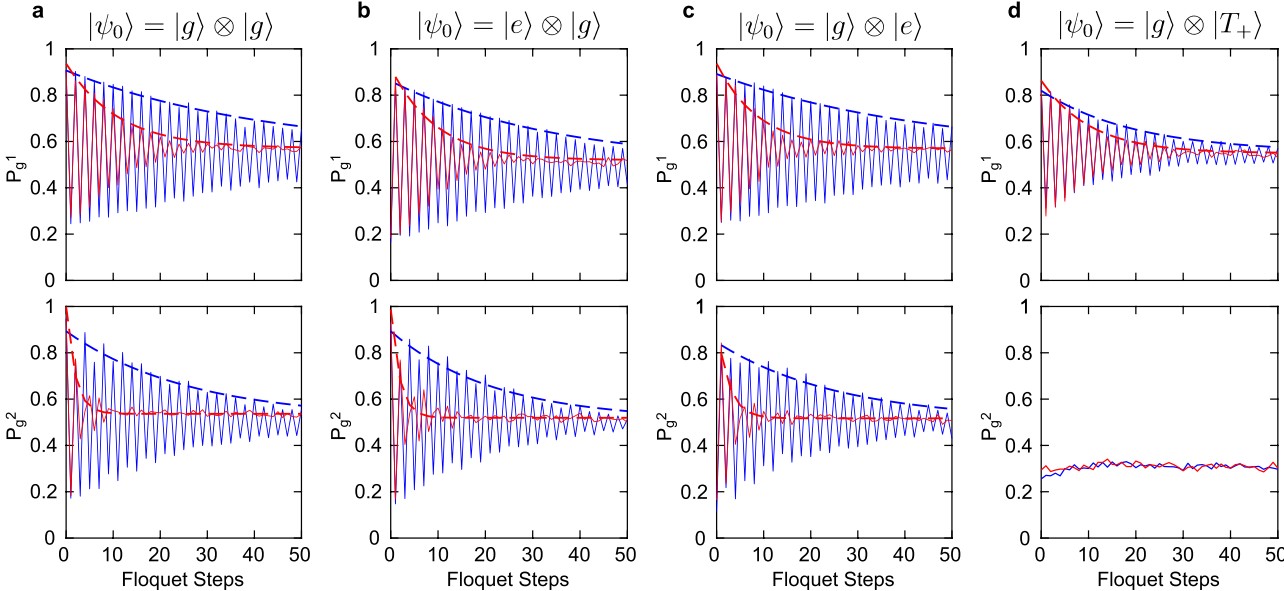

**Fig. 4 Floquet-enhanced spin swaps. a–d** Quality-factor enhancement of spin-eigenstate swaps for different initial states. In each figure, the top panel shows the measurements of ST qubit 1, and the bottom panel shows the measurements of ST qubit 2. The initial states are shown on the top, where $|g\rangle$ and $|e\rangle$ represent the ground state and the excited state of the ST qubit, respectively. The Floquet-enhanced $\pi$-pulse data are shown in blue, and the non-enhanced regular $\pi$-pulse data are shown in red. The fitted exponential decay envelopes are overlaid as dashed lines for all data except for the bottom panel in (**d**). The data are averaged over 4096 realizations.

We have now determined the optimal conditions for the Floquet enhancement. For the remainder of the paper, we set $J_1 = 270$ MHz and $J_3 = 150$ MHz with $t_1 = t_2 = 5$ ns for the SWAP operators $S_1$ and $S_2$, respectively, and we set $\tau = 1.4$ μs and $J_2 = 0.41$ MHz for the Ising interaction. To quantify the Floquet enhancement, we evolve the system for 50 Floquet steps and measure the ground-state return probabilities for both qubits after each step. The results are shown in Fig. 4a. Note that the system exhibits a clear subharmonic response to the Floquet operator. We extract a swap quality $Q$ by fitting the data with a decaying sinusoidal function $P_g(n) = \alpha \exp(-n/Q) \cos(n\pi) + \beta$, where $P_g(n)$ denotes the return probability at the $n$th Floquet step, and $Q$, $\alpha$, and $\beta$ are fit parameters. We also investigate the quality factor of the qubits under non-enhanced regular $\pi$ pulses. Here, we use the same interaction time $\tau = 1.4$ μs, but we turn off the interaction strength $J_2$ by setting the barrier gate pulse to zero. To further eliminate any effects associated with Floquet enhancement, we only apply $\pi$ pulses to one qubit, while the other qubit remains idle after initialization. Again, we apply 50 $\pi$ pulses and measure the ground-state return probability, and we fit the data with the same decaying sinusoidal function. By comparing the fit parameter $Q$, we can obtain the ratio between the quality factors of the qubits under Floquet-enhanced and non-enhanced $\pi$ rotations.

We find a ~3-fold quality-factor improvement on qubit 1, and ~9-fold improvement on qubit 2. The significant discrepancy between the quality-factor improvements of the two qubits is likely due to the large hyperfine gradient $\Delta_{34}$ in qubit 2, which causes an exceptionally low quality factor for non-enhanced $\pi$ rotations. The quality-factor enhancement is striking in this case. To extract an estimated uncertainty, we repeat the same experiment 30 times, and calculate the mean and the standard deviation of the quality-factor ratio, as shown in the first row of Table 1.

So far, we have initialized both ST qubits in their ground states. We can also initialize either ST qubit in its excited state by applying an extra $\pi$ pulse to the qubit immediately before the first

**Table 1 Quality-factor enhancements of both qubits for different initial states.**

| Initialization | Quality-factor enhancement | |
|---|---|---|
| | Qubit 1 | Qubit 2 |
| $|g\rangle \otimes |g\rangle$ | 3.60 ± 0.89 | 8.47 ± 3.29 |
| $|e\rangle \otimes |g\rangle$ | 3.24 ± 0.94 | 9.33 ± 2.96 |
| $|g\rangle \otimes |e\rangle$ | 3.15 ± 0.79 | 9.10 ± 2.87 |
| $|g\rangle \otimes |T_+\rangle$ | 1.92 ± 0.27 | N/A |

Here, $|g\rangle$ and $|e\rangle$ represent the ground state and the excited state of the ST qubit, respectively. Thirty sets of data are taken for each initialization, from which the means and the standard deviations are calculated.

Floquet step. We run the same experiment with different initial states and extract the quality factors by fitting the data (Fig. 4b, c). Again, for each initial state, we repeat the experiment 30 times and calculate the mean and the variance of the quality-factor ratio, which are listed in Table 1. The quality-factor improvements of both qubits are consistent across different initial states.

We also initialize the right pair as $|T_+\rangle = |\uparrow\uparrow\rangle$ and measure the quality-factor improvement on qubit 1 (Fig. 4d). We notice that the quality-factor ratio is much lower when the right pair is initialized in $|T_+\rangle$. This is not surprising since the effective Ising interaction between qubit 1 and qubit 2 (Eq. (2)) is only valid when both qubits are restricted to the $S_z = 0$ subspace. The reason why we still see a ~2-fold quality-factor improvement instead of no improvement at all is likely because of the imperfect $|T_+\rangle$ preparation due to thermal population of excited states[30]. Load errors will cause the right pair to occupy the ST-qubit ground or excited states a small fraction of the time. In these cases, the Floquet enhancement of the left-pair ST qubit is expected to occur. Thus, the overall quality factor should appear to improve slightly, because of the imperfect initialization. Correspondingly, it is likely that the imperfect initialization limits the quality-factor

enhancement when both qubits are initialized in ST-qubit eigenstates.

Finally, we emphasize that a Floquet drive, i.e., repeated SWAP gates, is required to realize the enhancement shown in Fig. 4. Based on the data of Fig. 4, the first SWAP gate is not substantially enhanced by the protocol. It is only subsequent SWAP gates that are enhanced. This is consistent with the requirement for a periodic drive in a DTC. As we discuss below, this periodic drive is also useful for constructing quantum gates.

## Discussion

Strictly speaking, a DTC only occurs in the thermodynamic limit[13]. Nonetheless, we argue the quality-factor enhancement we observe relies on the essential elements of DTC physics. The disordered Ising-coupled system in our device demonstrates a clear subharmonic response, as well as a robustness against pulse errors, both expected as defining signatures of the DTC. Our experiments also indicate the necessity of two essential ingredients for realizing the Floquet-enhanced $\pi$ pulses: (1) an effective Ising interaction, and (2) global $\pi$ pulses. If either of the components is missing, we no longer observe the significant quality-factor enhancement (Figs. 3 and 4d). These two components both ensure that the semiclassical dynamical decoupling can occur. In the thermodynamic limit, these components would ensure that eigenstates of the Floquet operator are long-range correlated, which is required for discrete time-translation symmetry breaking[5]. We have also shown that the quality-factor enhancement does not depend on the eigenstate into which either ST qubit is initialized (provided that the effective Ising coupling is maintained), which is another key feature of the DTC[13]. In the future, implementing these experiments in larger spin chains could lead to a verification that these effects in fact originate from the DTC phase.

We emphasize that we have observed Floquet enhancement associated with ST-qubit eigenstates undergoing $\pi$ pulses. In the language of single spins, we observed Floquet enhancement associated with swaps between spin eigenstates, when the total $z$ component of angular momentum for both spins vanishes. This observation is qualitatively consistent with expectations for qubits in a true many-body DTC, where the components of the qubits oriented along the direction defined by the Ising coupling are preserved[8]. While not a coherent SWAP gate, a spin-eigenstate swap (projection-SWAP), has significant potential to aid in readout for large qubit arrays[41].

The Floquet enhancement we observe can immediately be leveraged to perform additional quantum information processing tasks of significant importance. For example, recent theoretical work shows that entangled states of single spins (or superposition states of ST qubits) can be preserved, using Floquet operators identical to what we have demonstrated[16]. The same work also shows that single-qubit gates can be incorporated into this framework, and even two-qubit CZ gates can be implemented[16]. A significant potential advantage of these operations, compared with conventional single- and two-qubit gates, is that dynamical decoupling is an essential component of these operations, as discussed above.

As an illustration of the above capabilities, we perform simulations that show the preservation of the singlet state of an ST qubit by periodic driving under the evolution $U = S_{int} \cdot S_{12}$, where $S_{12}$ represents the execution of $S_1$ and $S_2$ in parallel. Figure 5a shows the return probability for the singlet state of an ST qubit defined on sites 3 and 4 of an $L = 6$ site spin chain. The ST qubits defined on the pairs of sites (1,2) and (5,6) are initialized to the product state $|\uparrow\downarrow\rangle$. When the interaction $J$ between neighboring ST qubits (the generalization of $J_2$ in Eq. (3)) is turned off, the

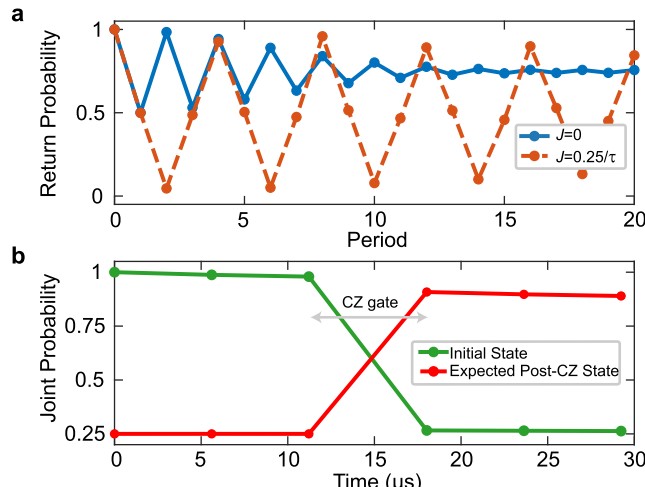

**Fig. 5 Preserving and generating entangled states. a** Return probability for the singlet state of an ST qubit defined on sites 3 and 4 of an $L = 6$ spin chain. The two remaining ST qubits are initialized in the state $|\uparrow\downarrow\rangle$. **b** Two-qubit probabilities before and after the execution of a two-qubit CZ gate, using the modified DTC protocol (for a chain of length $L = 4$). The initial state of ST qubit 1 is the triplet $|T_0\rangle$ and of qubit 2 is the singlet. The $x$ coordinate of each point is the total time of all the pulse sequences. The $y$ coordinate of each point is the joint two-qubit probability. The "expected post-CZ state" is an entangled state of the two ST qubits. The results in both panels are averaged over 4096 realizations.

maximum return probability decreases to ~0.75 at long times. In contrast, when $\tau J = 0.25$, with $\tau = 1.4\,\mu s$ as before, the maxima of the return probability remain higher than the noninteracting case out to 20 periods of evolution. In this case, the return probability shows a $4T$ periodicity, as the interactions produce a relative phase between the basis states $|\uparrow\downarrow\rangle$ and $|\downarrow\uparrow\rangle$ such that the original state is recovered only after four periods (after two periods this phase yields the $|T_0\rangle$ state, and the $|S\rangle$ return probability vanishes)[16]. We note that the calculated return probability $p = \mathrm{Tr}[\rho_{(3,4)}|S\rangle\langle S|]$ accounts for the fact that interactions will entangle the (3,4) pair with its neighbors through the use of the reduced density matrix $\rho_{(3,4)}$. The optimal value of $J$ is reduced by half compared to the previous simulations and experiments, due to the presence of two neighbors in the interior of the spin chain[34]. The possibility of stabilizing superposition states of ST qubits, such as singlets, also highlights the possibility of interleaving arbitrary single-qubit operations within this Floquet framework.

The condition $\tau J = 0.25$ also serves to implement a two-qubit CZ gate, for which simulations are shown in Fig. 5b for the $L = 4$ chain. Here, the ordinary DTC protocol with $\tau J = 0.5$ is applied for eight periods, followed by two periods with the reduced value $\tau J = 0.25$. Since the effective interaction between ST qubits is of Ising form, this yields a CZ gate up to single-qubit $z$ rotations: $CZ = e^{i\pi/4}e^{-i(\pi/4)\tilde{\sigma}_1^z}e^{-i(\pi/4)\tilde{\sigma}_2^z}e^{i(\pi/4)\tilde{\sigma}_1^z\tilde{\sigma}_2^z}$ (ref. [42]). Applying the necessary rotations by appropriately timed SWAP pulses during an additional evolution step with $t_{rot} = 4\,\mu s$ executes a full CZ gate[16], which we then preserve for another eight periods using the DTC protocol. Unlike the case of the Floquet-enhanced SWAP, here the gate itself is necessarily produced by the inter-ST-qubit coupling, and so it is not enhanced but rather enabled by it. Since the CZ gate and arbitrary single-qubit unitaries form a universal gate set, this suggests that the DTC-inspired methods presented here yield a promising direction for spin-based quantum computing.

The experimental investigation of all of these ideas remains an important subject of future work. We expect that these phenomena

can readily be explored in Si spin qubits. Barrier-controlled exchange coupling between Si spin qubits is now routine[43]. The operation of Si ST qubits in the regime where magnetic gradients exceed exchange couplings has also been demonstrated[15,44].

Note that to observe the Floquet enhancement, or to perform any of the protocols described in ref. [16], multiple $S_z = 0$ electron pairs undergoing the same Floquet operators are typically required. As we have shown above, one ST qubit alone cannot experience the Floquet enhancement without the other. This notion is consistent with expectations for many-body DTCs, which are true many-body phenomena. One can view the "extra" qubits undergoing repeated instances of the Floquet operator as the resource required to implement an improved operation on a specific qubit. It is also interesting to note that DTC-like behavior can emerge in systems with as few as two qubits in this nearest-neighbor-coupled system, as we have shown. Thus, only a relatively small number of qubits is required in order to realize the benefits of Floquet enhancement, highlighting its potential for further use in quantum information processing applications.

In summary, we have demonstrated Floquet-enhanced spin-eigenstate swaps in a four-spin two-qubit Ising chain in a quadruple quantum dot array. The system shows a subharmonic response to the driving frequency, and it also shows an improvement in swap quality factor even in the presence of pulse imperfections. We have also shown that the necessary conditions for this quality-factor enhancement are identical to some key components for realizing discrete time crystals. Our results also confirm the prediction of an effective Ising coupling that emerges between two exchange-coupled singlet–triplet qubits. This work indicates the possibility of realizing discrete time crystals using extended Heisenberg spin chains in semiconductor quantum dots, and suggests potential uses for discrete time crystals in quantum information processing applications.

## Methods

**Device.** The quadruple quantum dot device is fabricated on a GaAs/AlGaAs heterostructure substrate with three layers of overlapping Al confinement gates and a final Al top gate. The Al gates are patterned and deposited using E-beam lithography and thermal evaporation, and each layer is isolated from the other layers by a few nanometers of native oxide. The top gate covers the main device area and is grounded during the experiments. It likely smooths anomalies in the quantum dot potentials. The two-dimensional electron gas resides at the GaAs and AlGaAs interface, 91 nm below the semiconductor surface. The device is cooled in a dilution refrigerator with base temperature of ~10 mK. A 0.5-T external magnetic field is applied parallel to the device surface and perpendicular to the axis connecting the quantum dots.

**Pulse rise times.** The experimental values of $t_1 J_1^\pi$ and $t_2 J_3^\pi$ are much larger than 0.5, because the voltage pulses experienced by the qubits have rise times of ~1 ns. Supplementary Fig. 1 shows measured exchange oscillations for both ST qubits vs. evolution time. The observable frequency chirp at early evolution times demonstrates the effects of rise times in our system and shows that the π-pulse times yield $tJ > 0.5$.

**Simulation.** We simulate the Floquet-enhanced phenomena by evolving a four-spin array according to the Floquet operator $U = S_{\text{int}} \cdot S_2 \cdot S_1$, as defined in the main text. We set $t_1 = t_2 = 2$ ns, and $J_1^\pi = J_3^\pi = 250$ MHz for the SWAP operators $S_1$ and $S_2$ to give π pulses. While $t_1$ and $t_2$ are chosen to be 5 ns in the experiments, we expect the realistic SWAP times to be ~2–3 ns due to the pulse rise and fall times of ~1 ns. We include the π-pulse errors by adjusting the exchange couplings as $J_1 = J_1^\pi(1 + \epsilon)$ and $J_3 = J_3^\pi(1 + \epsilon)$, where $\epsilon$ represents the fractional error in the rotation angle of the π pulse applied to ST qubits 1 and 2.

For better comparison with the experimental data, the simulations take into account all known error sources, including state preparation, readout, charge noise, and hyperfine field noise. The initial state of each ST qubit is prepared as

$$\left| \psi_i \right\rangle = s_1 |g\rangle + s_2 |e\rangle + s_3 |T_+\rangle + s_4 |T_-\rangle , \qquad (6)$$

where $|g\rangle$ and $|e\rangle$ are the ground state and the excited state in the $\{ |\uparrow\downarrow\rangle, |\downarrow\uparrow\rangle \}$ basis. The exact spin orientation for the ground state is determined by the hyperfine gradient. The coefficient $|s_1|^2 = f_g$ represents the ground-state preparation fidelity, and we assume $|s_2|^2 = |s_3|^2 = |s_4|^2 = \frac{1}{3}(1 - f_g)$ for simplicity. We estimate $f_g$ to be

0.9 for ST qubit 1 and 0.95 for ST qubit 2 in our device. The preparation fidelity assumes errors from both the singlet loading and the charge separation. The readout errors are included by calculating the final ground-state return probability as

$$\bar{P}_g = (1 - r - 2q)P_g + r + q , \qquad (7)$$

where $P_g = |\langle g | \psi_f \rangle|^2$ is the true ground-state return probability. Here $r = 1 - \exp(-t_m/T_1)$ is the probability of the excited state relaxing to the ground state during measurements, with $t_m$ being the measurement time and $T_1$ being the relaxation time. Also $q = 1 - f_m$ is the probability of misidentifying the ground state as the excited state due to random noise. We set $t_m = 4$ μs, $T_1 = 60$ μs, and $f_m = 0.99$ for ST qubit 1, and $t_m = 6$ μs, $T_1 = 50$ μs, and $f_m = 0.95$ for ST qubit 2.

We use a Monte-Carlo method to incorporate charge noise and hyperfine field fluctuations. The values of the exchange couplings $J_i$ and the local hyperfine fields $B_i^z$ are randomly sampled from a normal distribution for each simulation run. We set the standard deviation for $J_i$ to be $J_i/(\sqrt{2\pi}Q)$, where $Q = 21$ is the exchange oscillation quality factor. We set the standard deviation for $B_i^z$ to be $\sigma_{B^z} = 18$ MHz, and we assume the mean values to be [0, 20, 0, 50] MHz plus a uniform magnetic field of 3.075 GHz (which accounts for the 0.5-T external magnetic field). The simulated data in Fig. 2 in the main text are obtained by averaging over 128 realizations.

The simulations of Fig. 5 do not include exchange coupling noise or state preparation and measurement errors. We neglected these errors to clearly illustrate the mechanisms underlying the singlet-state preservation and CZ gate. Hyperfine fluctuations with $\sigma_{B^z} = 18$ MHz were included, and the magnetic field values at the locations of each dot were 3.075 GHz, to account for the external magnetic field. To simulate the CZ gate of Fig. 5b in the main text, we simulate two periods of the Floquet operator $U = S_{\text{int}} \cdot S_{12}$, where $S_{12}$ represents the execution of $S_1$ and $S_2$ in parallel, with $\tau J = 0.25$. These two periods implement the CZ gate, up to single-qubit rotations. In the simulation, we evolve the system for two additional periods with the operator $U_{rot} = U_1 \otimes U_2$, where

$$U_1 = S_1 \cdot \exp\left(\frac{-i}{\hbar}\left[\frac{T_g - 2T_s - t_1^r}{2}\right]\frac{h}{2}\Delta_{12}\tilde{\sigma}_1^z\right) \cdot S_1 \cdot \exp\left(\frac{-i}{\hbar}\left[\frac{T_g - 2T_s + t_1^r}{2}\right]\frac{h}{2}\Delta_{12}\tilde{\sigma}_1^z\right)$$

$$U_2 = S_2 \cdot \exp\left(\frac{-i}{\hbar}\left[\frac{T_g - 2T_s - t_2^r}{2}\right]\frac{h}{2}\Delta_{34}\tilde{\sigma}_2^z\right) \cdot S_2 \cdot \exp\left(\frac{-i}{\hbar}\left[\frac{T_g - 2T_s + t_2^r}{2}\right]\frac{h}{2}\Delta_{34}\tilde{\sigma}_2^z\right)$$

where $T_g$ is the overall time of the operation ($U_{rot}$ lasts for a duration $T_g$), and $T_s$ is duration of the SWAP gate. We define the rotation time

$$t_1^r = \begin{cases} \pi/(2\Delta_{12}), & \text{if } \Delta_{12} > 0 \\ 3\pi/(2|\Delta_{12}|), & \text{if } \Delta_{12} < 0 \end{cases} \qquad (8)$$

and

$$t_2^r = \begin{cases} \pi/(2\Delta_{34}), & \text{if } \Delta_{34} > 0 \\ 3\pi/(2|\Delta_{34}|), & \text{if } \Delta_{34} < 0 \end{cases}. \qquad (9)$$

In total, $U_{rot}$ implements a π/2 rotation about the z-axis of both ST qubits via a spin-echo-like sequence, such that the overall operation over the four periods is an exact CZ gate[16].

To confirm that the behavior we report in the two ST qubits corresponds to that of an effective Ising spin chain, we also simulate an Ising spin chain. Define

$$H_I = \frac{h}{4}\sum_{i=1}^{N-1} J\sigma_i^z\sigma_{i+1}^z + \frac{h}{2}\sum_{i=1}^N B_i^z\sigma_i^z, \qquad (10)$$

and denote a Floquet operator

$$U(\tau) = \exp\left(-\frac{i}{\hbar}H_I\tau\right)$$
$$\times \prod_1^N \exp\left(-\frac{i}{\hbar}\left[(1+\epsilon)\frac{h}{2}J_\pi\sigma_i^x + \frac{h}{2}B_i^z\sigma_i^z\right]T_i^R\right), \qquad (11)$$

where $\epsilon$ is a pulse error, and $T_i^R = 1/(2\sqrt{J_\pi^2 + (B_i^z)^2})$, with $J_\pi = 250$ MHz. Supplementary Fig. 2 shows the results of simulations for an $N = 2$ Ising chain, after four Floquet steps, with $\tau = 1.4$ μs, $B_i^z = [20, 50]$ MHz, and $\sigma_{B^z} = \sqrt{2} \times 18$ MHz. These simulation conditions correspond to the data of Fig. 2 in the main text, and they agree with the data of that figure. This agreement provides additional strong evidence of the effective Ising coupling between ST qubits in our system.

Supplementary Fig. 3a shows the results of an $N = 8$ Ising spin chain after four Floquet steps, with $B_i^z = 20$ MHz. These results agree with the two-site data of Supplementary Fig. 2a and Fig. 2 in the main text, providing evidence that the behavior we observe in a two-qubit system corresponds to the expected behavior for larger systems. Supplementary Fig. 3b shows the simulated behavior after 1024 Floquet steps. The predicted semiclassical phase diagram discussed further below and in the main text is overlaid. The close agreement between the semiclassical phase diagram and the regions of state preservation provide additional confirmation of the link between the semiclassical phase diagram and the DTC phase.

**Semiclassical phase diagram calculation**. In ref. [10], Choi et al., explain the DTC-like behavior of their system with a semiclassical model. Inspired by their work, we present a related semiclassical model for our system. Let us consider an initial state of ST qubit 1: $|\psi_0\rangle = \cos(\theta_0/2)|g\rangle + e^{i\phi_0}\sin(\theta_0/2)|e\rangle$. Now, in the ideal case, we can imagine that after two Floquet steps, this initial state evolves to $|\psi_f\rangle = e^{-i\phi_1\sigma^z/2}e^{-i\theta\sigma^x/2}e^{-i\phi_2\sigma^z/2}e^{-i\theta\sigma^x/2}|\psi_0\rangle$. Here $\theta \approx \pi$ indicates a nominal $\pi$ pulse, $\phi_1 = 2\pi(J_2/2 + \Delta_{12} + \bar{B})\tau$, and $\phi_2 = 2\pi(-J_2/2 + \Delta_{12} + \bar{B})\tau$. The key assumption in this model is that the net effect of ST qubit 2 on ST qubit 1, is to generate the $\pi J_2\tau$ term in the propagator that switches sign after each Floquet step, because ST qubit 2 undergoes a nominal $\pi$ pulse. The change in sign of this part between Floquet steps is entirely a result of interactions in the system. Figure 3d of ref. [10] shows a single-qubit trajectory for this type of evolution. One can immediately see the relationship between this semiclassical approach and dynamical decoupling.

In order to see a period doubling in the system, even in the presence of errors, we require that $|\psi_f\rangle = |\psi_0\rangle$. In general, we can pick a $\theta_0$ and a $\phi_0$ to ensure that this is the case for a given $\theta$. To see a robust period doubling for $\theta \neq \pi$, we should see that approximately the same $|\psi_0\rangle$ is also unchanged under this evolution, even as we allow $\theta \neq \pi$.

To write down the actual evolution operator for our system, set

$$S_1 = \exp\left[-\frac{i}{\hbar}t_1\frac{h}{2}(\Delta_{12}\sigma^z + J_1\sigma^x)\right], \tag{12}$$

and let us define

$$S_{int}^- = \exp\left[-\frac{i}{\hbar}\tau\frac{h}{2}\left(\Delta_{12} + \bar{B} - \frac{J_2}{2}\right)\sigma^z\right] \tag{13}$$

$$S_{int}^+ = \exp\left[-\frac{i}{\hbar}\tau\frac{h}{2}\left(\Delta_{12} + \bar{B} + \frac{J_2}{2}\right)\sigma^z\right]. \tag{14}$$

In these definitions, we have suppressed the tildes, although the Pauli operators refer to the ST qubits. As before, $S_1$ describes a nominal $\pi$ pulse about the x-axis, and $S_{int}^-$ and $S_{int}^+$ describe the effect of interactions, depending on the state of ST qubit 2. The total Floquet operator over two steps is $U = S_{int}^+ S_1 S_{int}^- S_1$. To see a robust period doubling, we require that $|\psi_0\rangle = U|\psi_0\rangle$ for initial states with $\theta_0 \approx 0$.

We numerically calculate the eigenvectors of $U$ for the different interaction strengths and pulse errors $\epsilon$ we discuss in the manuscript. We will say that when the Floquet eigenstate $|\psi_0\rangle$ has a ground-state probability $P_g = |\langle g|\psi_0\rangle|^2 = \cos^2(\theta_0/2) > 0.9$, the system can enter the DTC-like phase. For each pulse error and $J_2$ configuration, we compute the eigenvectors for 256 different hyperfine and charge realizations. For each realization, we compute the value of $\cos^2(\theta_0/2)$, and then we average the values of $\cos^2(\theta_0/2)$ for all noise realizations for the same values of $J_2$ and pulse error. The phase diagrams obtained in this way are shown in Fig. 2 in the main text. To relate the phase diagram to our data in Fig. 2 in the main text, we rescale the values of the interqubit coupling $J_2$ we used in the simulation by 0.54/0.5, as discussed in the main text. The need for this correction occurs because of the error in our calibration of the interqubit coupling.

This construction clearly illustrates that without interactions or global $\pi$-pulses, the robust period doubling will not be observed, as discussed in ref. [10]. In this case, the eigenstates of $U$ are the same as the eigenstates of a single Floquet step, and there is no symmetry breaking. Without a global $\pi$-pulse, initial states with $\theta_0 \approx 0$ can only be approximately preserved after two Floquet steps (they are not exactly preserved), unlike the case with interactions, where these states are exactly preserved.

This semiclassical calculation is also valid for an end-spin of a longer spin chain, because we are considering a nearest-neighbor Ising chain. The argument we have provided applies to the first spin in the chain, and the interaction part depends on the state of the second spin. In our model, spin 2 is assumed to undergo perfect $\pi$ pulses. In a long spin chain, this assumption becomes more accurate, because one can view the effect of the third and first spins in the chain as stabilizing the $\pi$-rotations on spin 2, and so on.

## Data availability

The datasets generated during and/or analysed during the current study are available from the corresponding author on reasonable request.

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

## Acknowledgements

This research was sponsored by the Defense Advanced Research Projects Agency under Grant No. D18AC00025, the Army Research Office under Grant Nos. W911NF16-1-0260 and W911NF-19-1-0167, and the National Science Foundation under Grant Nos. DMR-1941673 and DMR-2003287. The views and conclusions contained in this document are those of the authors and should not be interpreted as representing the official policies, either expressed or implied, of the Army Research Office or the U.S. Government. The U.S. Government is authorized to reproduce and distribute reprints for Government purposes notwithstanding any copyright notation herein.

## Author contributions

J.VD., E.B., and J.M.N conceptualized the experiment. H.Q., Y.P.K., and J.VD. conducted the investigation. H.Q., Y.P.K., J.VD., E.B., and J.M.N. analyzed the data. S.F., G.C.G., and M.J.M. provided resources. All authors participated in writing. J.M.N. supervised the effort.

## Competing interests

The authors declare no competing interests.
