## [Peer Review File · Nature Communications]

REVIEWER COMMENTS

Reviewer #1 (Remarks to the Author):

The manuscript 'Floquet-enhanced spin swaps' is an interesting demonstration of exchange interaction in a four-spin system that makes connections to time crystals. The results are timely, but before recommending for publication, I would like to see some significant changes in the way this manuscript is written. In particular, this manuscript includes statements like 'four single-spin qubits' and 'multi-qubit processor', which raises expectations that are not demonstrated at all. This needs to be improved, which is perhaps best done by not mentioning qubits at all, since the focus is primarily on classical states, and to highlight the actual demonstration of four coupled spin states and the evolution of exchange oscillations by controlling the interaction strengths.

Comments:

- The abstract states: 'to boost the fidelity ...' However, no fidelity is stated in the main text.
- It is stated several times that a four single-spin qubit system is used, or different versions like 'multi-qubit processor'. However, no demonstration of four single spin qubits or a processor is reported.
- While the authors mention at several positions they only focus on a swap operation between eigenstates, it should be made more clear to the reader that there is no demonstration of a coherent swap and that the results themselves have therefore limited connection to quantum information transfer
- Please include in the supp or methods a section to show the rise times, now the main simply states on page 4: 'of about 1 ns'.
- Page 4: 'is either updown or downup'. If I am correct, this implies that Delta never changes sign and remains large. If this is the case, please include it to explain the statement. If it is not correct, then I do not understand why this statement is correct.
- Page 6. 'Furthermore ... qubits'. I believe only eigenstates are prepared.
- The S1 and S2 error figures are not explained very clear (and not even mentioned in the main tekst); e.g. how does the Pg1 probability relate to the S1 error?
- Page 6, the authors state that the enhancement is likely due to incorrect initialization. This is not explained very well. Why would an incorrect initialization not lead to only an off-set in the readout signal? And if it does not, what is the consequence for the other experiments?
- Can the authors add a discussion on the implications for coherent interactions? What is expected for the phase?

Reviewer #2 (Remarks to the Author):

The authors propose a method intended to improve the fidelity of a pair of swap gates when used simultaneously. The proposal is motivated by the physics of discrete time crystals. The authors find that a pair of swap gates acting on 2 separate pairs of qubits sees an improved fidelity if there is an additional coupling between the pairs. Only inputs 01 and 10 are considered for each pair of spins. While it is interesting to see DTC-like physics leading to an error correction in gates, the outcome in this work is extremely narrow in scope. For the reasons mentioned below, it can be assessed that the outcome is not sufficient to warrant publication in Nature Communications.

1) In order for the idea to work, the spins have to be in the Hilbert space with total $S_z=0$. If the total $S_z = +/-1$, the spin pairs are outside the singlet triplet space and the proposed enhancement will not work. In other words, the improvement in fidelity is applicable only if the input is 01 or 10 or linear combinations of these. If the qubits are in a general coherent superposition like $11 + 10$, it is not evident that the idea works. Indeed there can even be an increase in error due to the additional interactions when such a general input is given.

2) The idea works only if the swap gates have to be applied simultaneously on adjacent qubits pairs. As discussed in the conclusions, there is no improvement if the gates are not applied simultaneously. This makes the applicability of the idea very narrow.

3) While the experiments are interesting, nontrivial, and carefully carried out, most of the protocols mentioned in the manuscript, such as ST qubits, had been developed and previously published by the same group. It does not appear that the novelty of experimental methodology is a strength of the current report.

Reviewer #3 (Remarks to the Author):

This is an interesting paper which, inspired by the recently discovery of time crystals in Floquet systems, sees to improve SWAP operations in semiconductor qubits. I commend the authors for their curiosity and ambition.

At the heart of the paper is a story about two qubits which is encoded in their Floquet operator introduced in the section on Floquet-enhanced spin swaps. They wish to execute rotations by π about the x axis on two separate qubits but there are errors that will cause the angle and the direction to wander. Inspired by time crystals, they add a step in which the qubits undergo an Ising interaction. This "Floquet-enhancement" causes the approximate π flips to compose better - if performed an even number of times the result is a more accurate return to the starting state in the z-basis.

Three comments

1) While they are right to be inspired by time crystal physics, it is really a two qubit system and one really cannot invoke any of the understanding for macroscopic systems. For this reason I think the statements about long range entangled states in the supplement are not warranted. You really can't use $N \rightarrow \infty$ when discussing $N=2$.

2) Conversely, it should be possible to give intuition more specialized to the two qubit system, perhaps in a mean-field language married to spin echo ideas since the aim is to qualitatively understand their results and not a non-existent exactness in the limit of infinite time. I would suggest looking at Methods D of <https://arxiv.org/abs/1610.08057> for inspiration. I think they should look (computationally) at a two parameter family of choices of S_{int} - parameterized by the Ising interaction and field acting during that period. Is the interaction necessary or convenient?

3) I am not a good person to judge whether the Floquet enhancement is useful. It holds for two simultaneous swaps on the underlying physical qubits. Is this a useful primitive for quantum circuits? Should I be excited by it as a quantum engineer? It would be easier to judge the contribution of this paper to future of quantum dot based QIP if this was better discussed.

I cannot recommend the paper as is. Perhaps after responses to my three points, I can revisit the question. Basically, $N=2$ is not interesting for many body physics. So the authors need to make a strong case for (3) to justify space in this journal. Hopefully, other referees who are experts on that aspect are weighing in on this directly.

October 15, 2020

To the Reviewers:

Thank you for your insightful and constructive comments. We believe that the manuscript is significantly improved after incorporating the suggestions from all of the reviewers.

We have submitted a revised manuscript, and this letter details the changes we have made. We have addressed all of the comments from the reviewers. The reviewer comments are listed in blue text. Our responses are listed in black text. The manuscript is also marked up to show the changes we have made. At the end of this letter, we include a complete list of major changes to the manuscript.

Sincerely, on behalf of the authors,

John M. Nichol

Reviewer #1 (Remarks to the Author):

The manuscript ‘Floquet-enhanced spin swaps’ is an interesting demonstration of exchange interaction in a four-spin system that makes connections to time crystals. The results are timely, but before recommending for publication, I would like to see some significant changes in the way this manuscript is written. In particular, this manuscript includes statements like ‘four single-spin qubits’ and ‘multi-qubit processor’, which raises expectations that are not demonstrated at all. This needs to be improved, which is perhaps best done by not mentioning qubits at all, since the focus is primarily on classical states, and to highlight the actual demonstration of four coupled spin states and the evolution of exchange oscillations by controlling the interaction strengths.

We thank the reviewer for the positive appraisal of the manuscript.

Comments:

- The abstract states: ‘to boost the fidelity ...’ However, no fidelity is stated in the main text.

We have removed the term “fidelity” from the abstract.

- It is stated several times that a four single-spin qubit system is used, or different versions like ‘multi-qubit processor’. However, no demonstration of four single spin qubits or a processor is reported.

We have removed instances of the term “qubit” where they referred to the individual spins.

- While the authors mention at several positions they only focus on a swap operation between eigenstates, it should be made more clear to the reader that there is no demonstration of a coherent

swap and that the results themselves have therefore limited connection to quantum information transfer

We have added language to the first page of the paper emphasizing that the swap operation between spin eigenstates is not a coherent SWAP gate. In the spin-qubit literature, this is sometimes referred to as a projection-SWAP gate, and it is expected to be important for spin-state readout in extended chains.

- Please include in the supp or methods a section to show the rise times, now the main simply states on page 4: ‘of about 1 ns’.

We have replaced Supplementary Fig. 1 with data showing exchange oscillations in both singlet-triplet (ST) qubits as a function of evolution time. These data clearly illustrate the effects of rise times in our system.

- Page 4: ‘is either updown or downup’. If I am correct, this implies that Delta never changes sign and remains large. If this is the case, please include it to explain the statement. If it is not correct, then I do not understand why this statement is correct.

We meant to convey that during the course of the 8192 different realizations that are averaged together to produce a single pixel in the figures, the hyperfine gradients likely fluctuate, and the ground-state spin orientation of each ST qubit likely changes between $|\downarrow\uparrow\rangle$ or $|\uparrow\downarrow\rangle$, independent of the other ST qubit. We have clarified this in the main text.

- Page 6. ‘Furthermore ... qubits’. I believe only eigenstates are prepared.

We have changed the language to read “Furthermore, we have also shown that the quality-factor enhancement does not depend on the eigenstate into which either ST qubit is initialized.”

- The S1 and S2 error figures are not explained very clear (and not even mentioned in the main tekst); e.g. how does the Pg1 probability relate to the S1 error?

We have changed the notation, and we now use ϵ to refer to a pulse error. It is now defined at the end of the “Device and Hamiltonian” section.

- Page 6, the authors state that the enhancement is likely due to incorrect initialization. This is not explained very well. Why would an incorrect initialization not lead to only an off-set in the readout signal? And if it does not, what is the consequence for the other experiments?

When we attempt to prepare the right pair as a $|T_+\rangle$, it will actually be in the ground or excited state of the right ST qubit ($|\downarrow\uparrow\rangle$ or $|\uparrow\downarrow\rangle$) a small fraction of the time as a result of load errors. For these cases, we expect the Floquet enhancement for the left pair to occur. In the case where the right pair is properly initialized as a $|T_+\rangle$, we do not expect the Floquet enhancement for the left pair to occur, because the effective Ising coupling between ST qubits does not occur.

Thus, the overall oscillation of the left-pair ST qubit would be a weighted average of the non-enhanced and enhanced cases. One would therefore expect a resulting curve that resembles oscillations with weak enhancement.

These ideas are amplified by the additional discussion we have provided in response to Reviewer #3 (see below).

One potential consequence for the other experiments is that the Floquet enhancement of one pair is likely limited by the initialization fidelity of the other pair.

We have expanded the discussion of this point in the main text.

- Can the authors add a discussion on the implications for coherent interactions? What is expected for the phase?

A complete discussion of the implications for coherent interactions is beyond the scope of the current paper. However, we have posted a theory paper (arXiv: 2009.08469), which describes in great detail how this framework can be used for coherent operations. Specifically, we predict that the Floquet enhancement can be used to preserve entangled states, implement coherent rotations on single qubits, and implement CZ gates between ST qubits. We have summarized these findings in the Introduction and Discussion sections of the main text.

Reviewer #2 (Remarks to the Author):

The authors propose a method intended to improve the fidelity of a pair of swap gates when used simultaneously. The proposal is motivated by the physics of discrete time crystals. The authors find that a pair of swap gates acting on 2 separate pairs of qubits sees an improved fidelity if there is an additional coupling between the pairs. Only inputs 01 and 10 are considered for each pair of spins. While it is interesting to see DTC-like physics leading to an error correction in gates, the outcome in this work is extremely narrow in scope. For the reasons mentioned below, it can be assessed that the outcome is not sufficient to warrant publication in Nature Communications.

1) In order for the idea to work, the spins have to be in the Hilbert space with total $S_z=0$. If the total $S_z = +/-1$, the spin pairs are outside the singlet triplet space and the proposed enhancement will not work. In other words, the improvement in fidelity is applicable only if the input is 01 or 10 or linear combinations of these. If the qubits are in a general coherent superposition like $11 + 10$, it is not evident that the idea works. Indeed there can even be an increase in error due to the additional interactions when such a general input is given.

We agree that this physics requires neighboring pairs of spins to occupy a Hilbert space with $S_z = 0$, i.e., to be in a singlet-triplet (ST) subspace. However, given the significant theoretical and experimental research on ST qubits and related physics in quantum-dots (see e.g., Ref. [29] and citations to that reference), we expect that the new perspective offered by our work can be widely appreciated.

2) The idea works only if the swap gates have to be applied simultaneously on adjacent qubits pairs. As discussed in the conclusions, there is no improvement if the gates are not applied simultaneously. This makes the applicability of the idea very narrow.

We agree that the phenomenon we report is a collective phenomenon, because it requires all (both) qubits to undergo repeated operations, much as in a true many-body DTC. In the theoretical manuscript we have posted (arXiv: 2009.08469), we describe in great detail how this scenario can lead to improved single-qubit gates, multi-qubit gates, and entanglement preservation.

One can view the requirement for multiple $S_z = 0$ pairs undergoing repeated operations as the resource that enables the Floquet enhancement for certain operations. In general, one expects that improving quantum operations comes with some expense. For example, in the case of dynamical decoupling or dynamically corrected gates, the expense involves more complex pulses. In our work, the expense is another $S_z = 0$ pair that undergoes the same operation. This is a new perspective on quantum operations, and it takes the first steps toward understanding how many-body phenomena, like time crystals, can potentially be harnessed for quantum computing.

We have added text to clarify this in the Discussion section of the main text.

3) While the experiments are interesting, nontrivial, and carefully carried out, most of the protocols mentioned in the manuscript, such as ST qubits, had been developed and previously published by the same group. It does not appear that the novelty of experimental methodology is a strength of the current report.

We thank the referee for alerting us to the fact that we did not sufficiently emphasize the novelty of our experiments from the point of view of ST qubit research. To our knowledge, our work constitutes the first experimental demonstration of exchange-coupled ST qubits reported in the literature. Exchange-coupled ST qubits have been the focus of significant theoretical exploration over the last two decades, but they have not been implemented until now. Although we have not performed state tomography to fully characterize the two-qubit interaction, the appearance of the DTC-like phase diagrams is strong evidence of an effective Ising coupling between ST qubits. The emergence of an effective Ising coupling between exchange-coupled ST qubits is a long-standing prediction in this field, but this had not been experimentally observed prior to our work. We have added text to the abstract and introduction to emphasize these important points.

Reviewer #3 (Remarks to the Author):

This is an interesting paper which, inspired by the recently discovery of time crystals in Floquet systems, sees to improve SWAP operations in semiconductor qubits. I commend the authors for their curiosity and ambition.

At the heart of the paper is a story about two qubits which is encoded in their Floquet operator introduced in the section on Floquet-enhanced spin swaps. They wish to execute rotations by π about the x axis on two separate qubits but there are errors that will cause the angle and the

direction to wander. Inspired by time crystals, they add a step in which the qubits undergo an Ising interaction. This "Floquet-enhancement" causes the approximate π flips to compose better - if performed an even number of times the result is a more accurate return to the starting state in the z-basis.

Three comments

1) While they are right to be inspired by time crystal physics, it is really a two qubit system and one really cannot invoke any of the understanding for macroscopic systems. For this reason I think the statements about long range entangled states in the supplement are not warranted. You really can't use $N=\infty$ when discussing $N=2$.

We have deleted this section of the supplement.

2) Conversely, it should be possible to give intuition more specialized to the two qubit system, perhaps in a mean-field language married to spin echo ideas since the aim is to qualitatively understand their results and not a non-existent exactness in the limit of infinite time. I would suggest looking at Methods D of <https://arxiv.org/abs/1610.08057> for inspiration. I think they should look (computationally) at a two parameter family of choices of S_{int} - parameterized by the Ising interaction and field acting during that period. Is the interaction necessary or convenient?

We thank the reviewer for this insightful comment. We have performed an analysis inspired by the paper referenced above. In brief, our analysis considers an effective single-qubit evolution operator, which includes the effect of the neighboring qubit as a classical variable that alternates sign between Floquet steps. The resulting sequence of Floquet steps generates a series of rotations reminiscent of a dynamical decoupling pulse sequence. One can then compute for what values of the interaction strength and disorder eigenstates are exactly preserved by two Floquet steps. This analysis yields an effective phase diagram, which indicates regions of parameter space where the qubit eigenstates are also eigenstates of two periods of the Floquet operator.

We display these phase diagrams in Fig. 2 in the main text, and they agree with our data. The shape and size of the phase boundaries in the diagrams confirm that non-zero coupling between qubits significantly increases the range of pulse errors that can be tolerated.

Since the system under consideration is an effective nearest-neighbor Ising spin chain, unlike previous experiments on DTC physics, this calculation is identical to the semi-classical calculation one would make to identify the DTC phase diagram of an end-spin in a longer spin chain. In that case, one can view the effect of spin 3 (and also spin 1) as stabilizing the rotations of spin 2, which in turn create the dynamical decoupling effect on spin 1. Spins 2 and 4 stabilize the rotations on spin 3, and so on.

Thus, within this semiclassical picture, the phase diagrams we predict are identical to the true many-body DTC phase diagrams for a nearest neighbor Ising spin chain.

We have added simulations (Supplementary Figure 2), which show the simulated data for a two-spin Ising chain, which directly corresponds with our experiment. We have also added simulations

(Supplementary Figure 3) which show the expected results for an eight-spin Ising chain. Both sets of simulations show excellent agreement with our data after 4 instances of the Floquet operator. Moreover, the DTC-like regions in the eight-spin simulation after 1024 instances of the Floquet operator agree with the predicted phase diagrams from the analysis above.

We have added text describing this analysis in the main text on page 4.

3) I am not a good person to judge whether the Floquet enhancement is useful. It holds for two simultaneous swaps on the underlying physical qubits. Is this a useful primitive for quantum circuits? Should I be excited by it as a quantum engineer? It would be easier to judge the contribution of this paper to future of quantum dot based QIP if this was better discussed.

We thank the reviewer for this comment. We have posted a new theory paper (arXiv: 2009.08469), which details several different ways in which the Floquet enhancement and related effects could be useful. We show that this technique could be used to stabilize entangled states, perform single-qubit gates, and enact multi-qubit gates. Since the present experimental paper represents the first steps toward realizing this potential, we hope that it will excite quantum engineers. Although a full discussion of these topics is beyond the scope of the paper, we have added a summary to the main text.

We also emphasize, as above, that this work is the first experimental evidence of the effective Ising coupling that is predicted to emerge in exchange-coupled ST qubits. This is a significant advance for the spin-qubit community, and we have added text to emphasize this point.

I cannot recommend the paper as is. Perhaps after responses to my three points, I can revisit the question. Basically, $N=2$ is not interesting for many body physics. So the authors need to make a strong case for (3) to justify space in this journal. Hopefully, other referees who are experts on that aspect are weighing in on this directly.

We agree that our system is not a many-body system. Nonetheless, it is perhaps surprising, or at the very least encouraging for future applications, that at least some of the benefits of time-crystalline behavior can be realized in systems with as few as two qubits.

List of major changes

1. The term “fidelity” was removed from the abstract.
2. We have added a sentence to the abstract emphasizing that our results provide evidence for the predicted Ising coupling between exchange-coupled ST qubits.
3. All instances of the term “qubit,” where they referred to the individual spins, have been removed.
4. We have clarified that the spin-eigenstate swaps are not coherent SWAP gates in the last paragraph on the first page.
5. We have emphasized that our results are the first to confirm the Ising coupling between exchange-coupled ST qubits on the second page.
6. We have added a summary of the claims in our theory paper on the second page.
7. We have added the semiclassical phase diagrams to Fig. 2(a)-(b).

8. We have introduced the notation J_1^π and J_3^π to indicate the exchange coupling values that yield SWAP gates on the ST qubits.
9. We have introduced the symbol ϵ to indicate the pulse error.
10. We have added a paragraph describing the semiclassical analysis, suggested by Reviewer #3, on page 4.
11. We have clarified the effect of the fluctuating hyperfine gradient on page 5.
12. We have added a discussion about the semiclassical dynamical decoupling effect on page 5.
13. We have added a discussion about the implications of initialization errors on page 6.
14. We have added paragraphs in the Discussion section, which clarify the potential uses of our technique and its requirements.
15. We have added a figure and a section in the supplementary material describing the effect of pulse rise times in our system.
16. We have removed the discussion and figure related to global π pulses in large systems.
17. We have added a section and two figures on the semiclassical analysis and a comparison between our two-qubit system and an eight-qubit system.
18. Typos and font issues in the figures have been corrected.

REVIEWERS' COMMENTS

Reviewer #1 (Remarks to the Author):

The authors have addressed my comments. The manuscript is written in a more appropriate way and the conclusions are clear. I can therefore support publication, though the results are of limited impact given that the focus here is only on eigenstates.

Reviewer #2 (Remarks to the Author):

The authors have made substantial modifications to the manuscript and addressed the questions and comments from the referees. They have tried to argue that their work is of value in applications and have partly addressed some of our questions through another manuscript that they have published on arXiv. It is hard to convince that the attempt at Floquet enhancement does not hurt the fidelity of the gate when applied to more general initial states. Apart from this concern regarding the utility of the result, the manuscript seems technically all right. The work introduces a curious idea, inspired by the DTC physics, to build certain robust quantum gate operations but at the cost of an ST qubit. The manuscript can be recommended for publication if the following comments are addressed:

1) The use of the phrase Floquet enhancement gives the impression that there is an aspect of periodic driving here. This is not the case - the enhancement is observed in every gate instance of the swap operation.

2) The authors have added figure 5a which shows the return probability of a singlet pair after repeated application of certain unitary operations. It is argued that the return probability is improved upon adding a coupling to the spins surrounding the ST qubit. The plot conveys the message that the return probability after 20 cycles is enhanced, but further details of what is done here are not clear. It is not clear what precisely the U mentioned in line 413 is. I assume this is identical to the U introduced in Eqn 147.

In this case, it is not clear why the periodicity of the blue and the orange lines are different? Why are the minima of the return probability (troughs) occurring at different y -axis values in the two cases? In the presence of J , the time evolution of the interacting spin system will entangle the ST qubit with its neighbors; how is the return probability calculated for the ST qubit in this case?

3) The discussion of the CZ gate implementation in the lines following 431 is unclear. However, the authors appear to be merely pointing out that inter STqubit exchange coupling can also be implemented via an exchange coupling $tJ \sim 0.25$

The discussion gives the impression that the CZ gate undergoes a Floquet enhancement just like the swap gates do. This is a bit misleading.

The authors should clarify whether CZ gates benefit from a Floquet enhancement.

4) From the response to the last question of referee 1, it seems that the results in panels a and b of 5 are from simulations and not experiments. I could not figure this out easily from the text or the figure caption. Perhaps the authors could explicitly specify which results/plots in the manuscript are from experiments and which are from simulations.

Reviewer #3 (Remarks to the Author):

Of the three points I had raised, the authors have responded satisfactorily to the first two. On the third I must confess I don't quite know what to make of their response as I don't see a clear story about something like a universal gate set produced by Floquet techniques. They point to a second paper which I have not had the time to try to read with care. In any case, if the other referees - who presumably have greater familiarity with the state of quantum engineering - feel that the experimental manipulations achieved in this work are substantial enough to interest the broader community, I would not object to it being published.

Reviewer #2 (Remarks to the Author):

The authors have made substantial modifications to the manuscript and addressed the questions and comments from the referees. They have tried to argue that their work is of value in applications and have partly addressed some of our questions through another manuscript that they have published on arXiv. It is hard to convince that the attempt at Floquet enhancement does not hurt the fidelity of the gate when applied to more general initial states. Apart from this concern regarding the utility of the result, the manuscript seems technically all right. The work introduces a curious idea, inspired by the DTC physics, to build certain robust quantum gate operations but at the cost of an ST qubit. The manuscript can be recommended for publication if the following comments are addressed:

1) The use of the phrase Floquet enhancement gives the impression that there is an aspect of periodic driving here. This is not the case - the enhancement is observed in every gate instance of the swap operation.

Based on our data, the first SWAP gate is not substantially improved by the protocol. It is only subsequent SWAP gates that are improved. This is evident in Fig. 4. We have now clarified on page 6 that the enhancement is most pronounced after multiple SWAP gates, which is consistent with the DTC requirement for periodic driving, and that this notion is also useful for our gate construction.

2) The authors have added figure 5a which shows the return probability of a singlet pair after repeated application of certain unitary operations. It is argued that the return probability is improved upon adding a coupling to the spins surrounding the ST qubit. The plot conveys the message that the return probability after 20 cycles is enhanced, but further details of what is done here are not clear. It is not clear what precisely the U mentioned in line 413 is. I assume this is identical to the U introduced in Eqn 147. In this case, it is not clear why the periodicity of the blue and the orange lines are different? Why are the minima of the return probability (troughs) occurring at different y-axis values in the two cases? In the presence of J , the time evolution of the interacting spin system will entangle the ST qubit with its neighbors; how is the return probability calculated for the ST qubit in this case?

We have now clarified that the U in line 428 is the same as that in line 150, except that in 428, the intra-qubit swaps occur simultaneously.

The periodicity is $4T$ in the presence of inter-qubit interactions because these interactions generate additional phases between the up-down and down-up states that cause the initial singlet state to evolve to a triplet state after 2 periods. After 2 more periods, these phases unwind to restore the original singlet. For the orange curve in Fig. 5a, the minima are close to zero because the qubit is in the triplet state after 2 periods. This is not the case without inter-qubit interactions, and so the minima in the blue curve are higher.

To account for the entanglement between the ST qubit on sites 3 and 4 and its neighboring qubits, we use the reduced density matrix on sites 3 and 4 to define the singlet return probability.

Each of these points has now been clarified on page 7 of the revised manuscript.

3) The discussion of the CZ gate implementation in the lines following 431 is unclear. However, the authors appear to be merely pointing out that inter STqubit exchange coupling can also be implemented via an exchange coupling $t_J \sim 0.25$. The discussion gives the impression that the CZ gate undergoes a Floquet enhancement just like the swap gates do. This is a bit misleading. The authors should clarify whether CZ gates benefit from a Floquet enhancement.

The referee is correct that the CZ gate is not enhanced by the Floquet driving. The point we are making here is that a CZ gate can easily be embedded into the Floquet driving protocol by simply reducing the inter-qubit exchange coupling over 2 periods. Our simulations (Fig. 5b) show that the application of this gate does not disrupt the state protection afforded by the Floquet driving. Since the CZ gate is a maximally entangling two-qubit gate, this provides evidence that a universal gate set can be integrated into the Floquet driving protocol so that gate operations can be performed while states are protected. This idea is investigated further in Ref. [16].

We have now added additional discussion in the Supplementary Material and on page 7 in the main text to provide more details and to avoid this possible point of confusion.

4) From the response to the last question of referee 1, it seems that the results in panels a and b of 5 are from simulations and not experiments. I could not figure this out easily from the text or the figure caption. Perhaps the authors could explicitly specify which results/plots in the manuscript are from experiments and which are from simulations.

We have now made it clear on page 7 that the results in Fig. 5 are from simulations and not experiments.

Reviewer #3 (Remarks to the Author):

Of the three points I had raised, the authors have responded satisfactorily to the first two. On the third I must confess I don't quite know what to make of their response as I don't see a clear story about something like a universal gate set produced by Floquet techniques. They point to a second paper which I have not had the time to try to read with care. In any case, if the other referees - who presumably have greater familiarity with the state of quantum engineering - feel that the experimental manipulations achieved in this work are substantial enough to interest the broader community, I would not object to it being published.

In Fig. 5, we show evidence from numerical simulations that a maximally entangling CZ gate can be embedded in the Floquet driving sequence without disturbing the state protection afforded by this sequence. This gate is performed simply by reducing the inter-qubit exchange coupling for two periods. Similar results hold for single-qubit operations as well (Ref. [16]). Collectively, these results

show that a universal gate set can be integrated into the Floquet sequence. In summary, Floquet techniques enhance single-qubit spin -eigenstate operations and preserve quantum states, and they are compatible with a universal gate set based on the CZ gate.